# Diagnostic Role of Native T1 Mapping Compared to Conventional Magnetic Resonance Techniques in Cardiac Disease in a Real-Life Cohort

**DOI:** 10.3390/diagnostics13142461

**Published:** 2023-07-24

**Authors:** Giovanni Donato Aquaro, Silvia Monastero, Giancarlo Todiere, Andrea Barison, Carmelo De Gori, Crysanthos Grigoratos, Maria Luisa Parisella, Lorenzo Faggioni, Dania Cioni, Riccardo Lencioni, Emanuele Neri

**Affiliations:** 1Academic Radiology Unit, Department of Surgical, Medical and Molecular Pathology and Critical Area, University of Pisa, 56126 Pisa, Italy; 2Gabriele Monasterio CNR-Tuscany Foundation, 56127 Pisa, Italy; silvia.monastero@gmail.com (S.M.);; 3Academic Radiology Unit, Department of Translational Research and of New Technology in Medicine and Surgery, University of Pisa, 56126 Pisa, Italy

**Keywords:** cardiac magnetic resonance, T1 mapping, late gadolinium enhancement, T2-STIR

## Abstract

We sought to compare native T1 mapping to conventional late gadolinium enhancement (LGE) and T2-STIR techniques in a cohort of consecutive patients undergoing cardiac MRI (CMR). CMR was performed in 323 patients, 206 males (64%), mean age 54 ± 8 years, and in 27 age- and sex- matched healthy controls. In T2-STIR images, myocardial hyperintensity suggesting edema was found in 41 patients (27%). LGE images were positive in 206 patients (64%). T1 mapping was abnormal in 171 (49%). In 206 patients (64%), a matching between LGE and native T1 was found. T1 was abnormal in 32 out of 41 (78%) with edema in T2-STIR images. Overall, LGE and/or T2-STIR were abnormal in 209 patients, whereas native T1 was abnormal in 154 (52%). Conventional techniques and T1 mapping were concordant in 208 patients (64%). In 39 patients, T1 mapping was positive despite negative conventional techniques (12%). T1 mapping was able in conditions with diffuse myocardial damage such as cardiac amyloidosis, scleroderma, and Fabry disease (additive role in 42%). In contrast, T1 mapping was less effective in cardiac disease with regional distribution of myocardial damage such as myocardial infarction, HCM, and myocarditis. In conclusion, conventional LGE/T2-STIR and T1 mapping are complementary techniques and should be used together in every CMR examination.

## 1. Introduction

Cardiovascular magnetic resonance (CMR) plays an increasingly crucial role in diagnosing cardiac diseases, serving as the gold standard imaging technique for assessing cardiac function and accurately measuring ventricular volumes and mass. What sets CMR apart is its unique ability to perform soft-tissue characterization [1]. CMR employs various conventional techniques such as T1 fast-spin echo (FSE) with/without fat saturation, T2 short tau inversion recovery (STIR) pulse sequence, and late gadolinium enhancement (LGE) techniques to evaluate myocardial tissue. These techniques are effective in detecting fat infiltration/metaplasia, edema, and fibrosis, respectively. They have proven to be robust and reproducible. The integration of findings from these conventional techniques with morphological and functional features enables comprehensive diagnosis of myocardial diseases.

Certain features obtained by this technique, such as the presence, pattern of presentation, and extent of fibrosis, have been demonstrated to play a significant prognostic role in various cardiac conditions [2]. However, a notable limitation of these conventional techniques, particularly LGE and T2-STIR, is their qualitative or semiquantitative nature, allowing only a comparative analysis (hypointense, isointense, hyperintense) between normal and diseased myocardium.

The T1 mapping technique may overcome these limitations by measuring the intrinsic myocardial T1 relaxation time on a pixel-wise basis [3]. This means that every pixel on a T1 map represents an absolute T1 value, allowing for precise quantification. Abnormalities in myocardial T1, whether they are global or regional, are determined by comparing them to reference values expressed in milliseconds [4]. Consequently, T1 mapping has the potential to detect diffuse structural changes in the myocardium that may not be assessable by other non-invasive techniques, including LGE. This capability makes T1 mapping a valuable tool for identifying and characterizing subtle myocardial alterations that could have gone unnoticed using conventional imaging methods.

Non-contrast myocardial T1, often referred to as “native” T1, is a term used to distinguish it from post-contrast T1. The most important biological determinants of an increase in “native” T1 are edema (increase in tissue water in, i.e., acute infarction, inflammation, or acute toxic damage), the increase in interstitial space for fibrosis (infarction, chronic myocarditis, cardiomyopathy, etc.), or for amyloidotic proteins deposition [5]. In contrast, the two most important causes of low native T1 values are lipid overload (i.e., Anderson–Fabry disease, lipomatous metaplasia) and iron overload [6].

T1 mapping techniques may be performed using two different families of pulse sequences: those based on inversion recovery (IR), including the standard Look-Locker sequence, the MOdified Look-Locker inversion recovery (MOLLI) sequence, and the shortened MOLLI (ShMOLLI) sequence; and those, less frequently used, based on a saturation recovery such as the saturation recovery single-shot acquisition (SASHA) and the saturation-pulse prepared heart-rate-independent inversion recovery (SAPPHIRE) [5]. The MOLLI technique is the most frequently used. It uses a steady-state free precession (SSFP) readout that drives the IR to recover more quickly and reaches a steady state that is less than the equilibrium magnetization (M0). The effect of the readout is an apparent recovery time referred to as T1*, which is less than the actual longitudinal recovery time, T1. The MOLLI method samples the IR curve at multiple inversion times using single-shot imaging spaced at heart beat intervals. Multiple inversions are used with different trigger delays in order to acquire measurements at different inversion times to sample the IR curve more evenly. Recovery periods are needed between the inversions to ensure that samples from the different inversions are from the same recovery curve. The first method for MOLLI described is the 3(3)3(3)5, where number in parenthesis represents the number of heart beats for recovery of magnetization and the others the numbers of images acquired in different heart beats following a single IR pulse. Nowadays, the most used approach is the 5(3)3. A number of protocol modifications (such as the ShMOLLI technique) have been proposed to shorten the acquisition duration or to improve the accuracy or precision [4].

Post-processing methods play a crucial role in T1 mapping. To begin, the signal intensity of each voxel or group of voxels is measured in every image acquired at different inversion times [7]. These signal intensity values are then plotted against the inversion time, resulting in a curve. In IR-based techniques, the initial part of the curve exhibits a descending pattern until reaching the null point. To transform the curve into a negative exponential curve, this descending portion is inverted by assigning negative values. Next, a mathematical fitting process is applied to the curve. By fitting a mathematical equation to the curve, the T1 value is derived. This equation describes the behavior of the curve and allows for the calculation of T1 relaxation time. Finally, a gray-scale T1 map is generated by assigning a signal value corresponding to the respective T1 value at each voxel or group of voxels [3]. In visualization, these T1 maps are typically presented as parametric color maps, which can be displayed on workstations. The color map allows for a visual representation of T1 values, aiding in the interpretation and identification of variations in tissue characteristics.

The T1 mapping technique has some limitations: (1) the variability due to factors such as heart rate, motion artifacts, arrhythmias, and imaging parameters and, mostly, magnetic field inhomogeneity; (2) the lack of standardized reference values for myocardial T1 in the myocardium for which each CMR laboratory must create its own reference values; (3) in many cardiac conditions, different phenomena may coexist such as fibrosis, edema, inflammation, and even opposite phenomena (as fat and fibrosis) that may affect the myocardial T1 relaxation, leading to false-positive or false-negative findings; (4) the additional time required for T1 mapping sequences can lead to longer scan durations, while SCMR position documents suggest to only acquire three ventricular short-axis views for T1 mapping; (5) the differences in acquisition schemes have a direct effect on the range of normal and abnormal T1 with a given technique, which means that absolute T1 values can only be directly compared when they are obtained with the same acquisition scheme at the same field strength and using the same post-processing methods [5].

The aim of the present study was to compare the diagnostic role of conventional CMR techniques to native T1 mapping in a real-life cohort of non-selected consecutive patients undergoing CMR.

## 2. Materials and Methods

A total of 340 consecutive patients who underwent CMR for clinically suspect of cardiac diseases from August 2019 to January 2020, and 27 healthy controls were included in this study. We excluded patients with congenital heart disease (T1 mapping was not included in clinical protocol) and those with cardiac tumors. Finally, after CMR patients with sub-optimal images were excluded, those with contraindication for gadolinium-based agents and those that did not complete the CMR exam were excluded. The final population included 323 patients (17 excluded for sub-optimal image quality or for incomplete exam). The study was conducted in accordance with the Declaration of Helsinki, and approved by the Ethics Committee of the Area Vasta Nord-Ovest (protocol code 0016706 of 26 September 2018). Informed consent was obtained from all subjects involved in the study.

### 2.1. CMR Protocol

All CMR exams were performed on a 1.5 T scanner (Artist, GE Healthcare, Milwaukee, WI, USA) using dedicated cardiac phased-array surface receiver coil, and vectorcardiogram triggering. According to the protocols recommended by the Society for Cardiovascular Magnetic Resonance, we acquired cine steady-state free precession (cine-SSFP) images, T1-FSE with and without Fat-saturation, T2-STIR, and LGE at 10 min after gadolinium injection in the short-axis (9–13 images covering the entire LV), 2-chamber, and 4-chamber planes. Short axis cine-SSFP images were acquired immediately after gadolinium injection for hyperemia assessment. The same protocol was repeated at CMR-II. According to the protocols recommended by the Society for Cardiovascular Magnetic Resonance [8], we acquired T1 mapping sequence using a MOdified Look-Locker Inversion Recovery (MOLLI) method with 5(3)2 protocol where 5 images are acquired in consecutive heart beats after the first IR, then a recovery period of 3 heart beats intervals, and, finally, a new IR is followed by the acquisition of 3 other images.

We obtained three parallel short-axis slices, including the base, mid cavity, and apex of the left ventricle, at the same cardiac phase (end diastole) [8].

### 2.2. CMR Post-Processing

All CMR studies were analyzed offline using a workstation with dedicated cardiac software (cvi42, Circle Cardiovascular Imaging, Calgary, Alberta, Canada) and consensus was obtained among three experienced observers who were blinded to the clinical presentation results. To evaluate the LV global and regional function and calculate the LV mass, the endocardial and epicardial borders were manually drawn in the end-diastolic and end-systolic short-axis cine SSFP images. Papillary muscles and trabeculations were not included in the myocardium. LV end-diastolic volume (EDV), LV end-systolic volume, EF, and LV mass were determined.

In T2-weighted images, edema was evaluated visually and confirmed if the ratio of signal intensity (SI) between the myocardium and the mean SI of the skeletal muscle was ≥2. The extent, diffuse, or focal, and the pattern of distribution of edema, ischemic-like (transmural or subendocardial), or non-ischemic (sub-epicardial, mid-wall) features were evaluated.

LGE was qualitatively evaluated as for edema: the ischemic pattern (subendocardial/transmural, confluent scar in a territory of distribution of one coronary artery) or non-ischemic pattern of distribution (i.e., subepicardial or mid-ventricular enhancement, not limited to a coronary artery territory) was recorded. The presence of LGE was evaluated measured using a previously validated method. Criteria for CMR diagnosis of different cardiac conditions are shown in Table 1 and Table 2 [9,10,11,12,13,14,15,16,17,18,19,20,21,22,23,24].

Absolute T1 values were directly compared with local sex- and age-referenced range obtained using the same acquisition scheme with the same MRI machine strength and pulse sequence parameters and the same post-processing method [25]. To measure average myocardial T1 times, epicardial and endocardial contours of LV were manually traced. The average value of all the global LV myocardium was measured. By the visual assessment of the obtained T1 maps, regions of interest (ROIs) were also traced in myocardial areas with abnormal native T1 and the respective T1 value measured. In the presence of an abnormal T1 value, the abnormalities were classified as: (1) increased or decreased native T1; (2) diffuse or focal T1 abnormalities [6].

### 2.3. Statistical Analysis

Values are presented as the mean ± SD or as the median and interquartile range (IQR) for variables with normal and non-normal distributions, respectively. Values with non-normal distribution according to the Kolmogorov–Smirnov test were logarithmically transformed for parametric analysis. Qualitative data are expressed as percentages. Categorical variables were compared using the Chi-square test or Fisher’s exact test when appropriate. Continuous variables were compared by the Student’s independent *t*-test and ANOVA or by the Wilcoxon non-parametric test when appropriate. A *p*-value lower than 0.05 was considered statistically significant.

## 3. Results

The final population included 323 patients, 206 males (64%), mean age 54 ± 18 years. The baseline characteristics are shown in Table 3. Briefly, the main indications for CMR were: suspect of arrhythmogenic cardiomyopathy (ARC) in 20% of cases; non-ischemic dilated cardiomyopathy (DCM) in 19%; hypertrophic cardiomyopathy (HCM) in 16%; chest pain without obstructive coronary artery in 14% of patients; other indications (amyloidosis, scleroderma, previous myocardial infarction, pericarditis, LV non-compaction, etc.) in the remaining cases. We also included 27 (8%) of age- and sex- matched healthy controls. As evident in Table 4, conventional T2-STIR images that were acquired in 154 patients show myocardial hyperintensity compatible for edema in 41 patients (27%) of patients. Myocardial fat infiltration/metaplasia in T1-FSE with/without fat saturation is found in 20 patients, all of them having a positive non-ischemic LGE. LGE is positive in 198 (61%) of patients: an ischemic pattern of distribution was found in 15 (5%), non-ischemic in 183 (57%), including those with a specific LGE pattern for cardiac amyloidosis (12 patients, 4%).

In T1 mapping, the average native T1 is 1040 ± 55 ms. A total of 65 (20%) patients show a global increase in native T1 (mean T1 1232 ± 85 ms), while it decreases in 11 (3%) patients (mean T1 885 ± 85 ms). A regional increase in native T1 is found in 160 (46%) patients, 52 of them with global increase in T1 values.

### 3.1. Conventional CMR vs. T1 Mapping

Among the 41 patients with signs of myocardial edema at T2-STIR, 32 (78%) also have an increase in regional native T1; in 9 patients, edema is not associated with a consistent increase in T1. Overall, a matching between T2-STIR and native T1 is found in 185 (57%) patients.

T1 is normal in 74 (36%) patients with positive LGE and abnormal in 33 with negative LGE. Overall, a perfect matching between these two techniques is found in 206 (64%) patients (both positive in 132, both negative in 74 patients). No significant differences are found for the pattern of LGE: native T1 is increased in 118 out of 186 patients (63%) with non-ischemic LGE vs. 14 out of 20 (65%) with ischemic pattern (*p* = 0.81). Native T1 is normal in 2 out of 10 patients without gadolinium injection.

Among the 33 patients with abnormal T1 and negative LGE, only 4 (12%) have edema in T2-STIR images

### 3.2. Diagnostic Role of T1 Mapping

Overall, the CMR findings confirm the initial suspicion in 149 patients (50%), yielded an alternative diagnosis in 41 (14%), are non-specific (not allowing a definitive diagnosis) in 78 (26%), and completely negative in 28 (10%) (Table 5).

Conventional CMR techniques (LGE and/or T1-FSE and/or T2-STIR) are positive in 209 patients (71%), while native T1 is abnormal in 154 (52%) of patients (*p* < 0.0001). As evident in Figure 1, mapping and conventional techniques are concordant in 208 patients (64%); in 76 patients (24%), conventional imaging is positive in the presence of normal T1 values; in 39 patients, T1 values are abnormal despite negative findings in conventional CMR (12%).

Interestingly, in the suspicion of myocarditis, a concordance between conventional techniques and T1 mapping is found in 50% of patients, whereas all the remaining patients have positive conventional CMR but negative T1 mapping (Figure 2). In patients with myocarditis, conventional techniques are more frequently positive than T1 mapping (*p* < 0.0001). Similar results are found in the case of myocardial infarction, where a concordance appears in 79% of patients, whereas positive conventional techniques with negative mapping is apparent in 21% of patients (Figure 2). In both of these two conditions, mapping has no additive role over conventional techniques.

On the contrary, opposite results are found about amyloidosis and scleroderma where native T1 is abnormal in all the patients (Figure 2). Particularly abnormal T1 mapping is the only finding in 17% and 47% of patients, respectively. T1 mapping is more frequently positive in scleroderma than conventional techniques (*p* < 0.0001).

Abnormal T1 mapping is the only CMR abnormality in 11% of DCM, in 3% of HCM, 16% of ARC, and in 18% of other conditions (Figure 3). On contrast, native T1 is within the range of normality despite a positive LGE and/or T2-STIR in 22% of DCM, 40% of HCM, 36% of ARC, and in 28% of other conditions.

In HCM, myocardial abnormalities are more frequently detected using conventional techniques than with T1 mapping (*p* = 0.0006).

### 3.3. Diffuse vs. Regional Myocardial Damage

T1 mapping is able to give more additive information than conventional techniques in cardiac diseases with diffuse myocardial damage such as scleroderma, amyloidosis, and Fabry disease. In these conditions, T1 mapping is abnormal in 24 out of 24 patients, demonstrating an additive role over the conventional techniques in 10 patients (42%).

In contrast, T1 mapping is less effective in cardiac conditions with a regional\segmental distribution of myocardial damage such as myocardial infarction, myocarditis, and HCM. In these conditions, T1 is abnormal in 52 out of 88 patients (58%) but only demonstrated an additive role in one of them (1%). Conventional techniques have an additive role in 36 patients (41%).

An explanation for the low effectiveness of T1 mapping in regional myocardial damage could refer to the limited coverage of LV myocardium. Indeed, T1 mapping is acquired in three short-axis slices, covering an average of 17 ± 4% of LV mass.

## 4. Discussion

In the present study, we evaluated the clinical impact of T1 mapping as an additive imaging tool in a cohort of non-selected patients undergoing CMR. The main results may be summarized as follows: (1) conventional CMR for tissue characterization (LGE, T1-FSE, and T2-STIR) and T1 mapping are complementary techniques in most of the cardiac conditions: they provide concordant findings in 64% of patients; (2) native T1 mapping has an additive diagnostic role over conventional techniques in a range of 6–12% of cases; (3) in 24% of cases, conventional CMR techniques detect myocardial abnormalities despite normal native T1; (4) the role of T1 mapping is different in cardiac diseases: we find that T1 mapping is superior to conventional approaches in cardiac conditions such as amyloidosis, Fabry, and scleroderma, characterized by a diffuse myocardial involvement; (5) conventional techniques are superior to mapping for the evaluation of myocardial disease with segmental distribution such as myocardial infarction, hypertrophic cardiomyopathy, and myocarditis.

T1 mapping is a quantitative CMR technique allowing voxel-wise quantification of myocardial native T1 [4]. Native T1 may be abnormal in different cardiac conditions based on the changes in myocardial content of water, proteins, and fat [5]. Water is characterized by the greatest T1 values because of a fast “tumbling” rate of small, rapidly rotating molecules [1]. Then, myocardial T1 is mostly increased in the presence of augment of free-water content such as in myocardial edema. Myocardial edema may be found in myocarditis, acute/subacute myocardial infarction, and in every case of recent myocardial damage [5,8,9]. Indeed, it was described also in hypertrophic cardiomyopathy, in cocaine-induced myocardial damage, in scleroderma, and in many other conditions [1,10].

Increased native T1 may also be found in myocardial fibrosis because of augmented water content of interstitial space enlarged by the collagen matrix of scar [4].

Myocardial T1 is also severely increased in amyloid deposition [5,11], whereas it is decreased in the presence of intramyocardial fat infiltration or in Fabry disease because of lysosomal sphingolipids accumulation [12,13].

By these premises, T1 mapping and conventional techniques, such as LGE and T2-STIR, are overlapping in many cardiac conditions, as confirmed by the results of the present study, where the concordance between these techniques is seen in 64% of patients. However, T1 mapping has some advantage over conventional techniques. Being a quantitative technique, T1 mapping may allow for the identification of myocardial disease characterized by a diffuse and homogeneous damage, as evident in early stages of DCM, amyloidosis, Fabry disease, and in scleroderma [5,11,12,13,14,15].

In the matter of fact, T1 mapping demonstrates an additive diagnostic role in 12% of patients who present a negative LGE and T2-STIR but abnormal native T1.

We find that the value of T1 mapping is particularly relevant in some conditions such as cardiac amyloidosis where it is positive in 17% more patients than LGE. In cardiac amyloidosis, LGE has a very specific pattern, characterized by a diffuse subendocardial enhancement, an early darkening of signal of blood cavity, and a null defect of myocardium. The specificity of this pattern is near to 100% but it may be absent in the early stage of this disease [11]. Amyloid deposit is associated with a great increase in myocardial native T1, also in the early stages. Then, the presence of diffuse increase in native T1, summed to concentric hypertrophy and other morphological signs, such as thickening of atrial septal walls and pericardial effusion, and associated with clinical presentation, may allow for a diagnosis of amyloidosis, even in the absence of the specific LGE pattern.

Scleroderma is usually associated with edema and microscopic fibrosis (Figure 4). Both of them may present with a diffuse, non-regional distribution. Microscopic fibrosis could be not detected by LGE [5,23]. The identification of diffuse edema is very challenging using conventional, qualitative, T2-STIR pulse sequence, because of a lack of comparison with “normal” myocardium. In this setting, native T1 and ECV evaluation could be very useful. Native T1 is abnormal in all the patients with these conditions and an additive role of T1 mapping over conventional techniques is found in 47% of cases of scleroderma. T1 is found to be positive in a significantly higher percentage of cases of scleroderma than LGE\T2-STIR.

T1 mapping is also instrumental for the diagnosis of the case of Fabry disease where conventional techniques are completely normal (Figure 5), with only a mild concentric hypertrophy but areas of low T1 detected. In this case, alpha-galactosidase test confirmed the diagnosis [18,19,20].

An advantage of T1 mapping is the absence of contrast injection that is required for LGE technique. The nephrogenic systemic sclerosis, a rare complication of gadolinium-based contrast agents, is associated with severe kidney disease. Contrast injection in patients can induce a severe reduction in glomerular filtration rate, which is potentially dangerous. The lack of LGE images make CMR less effective in ruling out cardiac disease and this is particularly relevant in subjects with ventricular arrhythmias (such as frequent PVC) with normal cardiac structure and function. In such conditions, the identification of myocardial fibrosis is very important because the prognosis depends on the presence of a structural myocardial disease [5,16]. T1 mapping may help to rule out structural disease when LGE is not acquired because of renal condition or in the case of refusal of injection by the patients.

In contrast, T1 mapping results in a less effective ability to identify abnormalities in conditions such as myocarditis, small myocardial infarction, or in the case of non-ischemic fibrosis. Indeed, abnormal native T1 is found in 50% of patients with myocarditis, whereas LGE and/or T2-STIR are positive in all of these patients. Similarly, T1 native is abnormal in 71% of acute\chronic myocardial infarction patients, while LGE is positive in all of these patients and T2-STIR in all of the patients with acute myocardial infarction.

A possible explanation of these findings may be found in some technical aspects of T1 mapping acquisition. T1 mapping images usually do not cover the entire left ventricle but the SCMR position paper [8] suggests acquiring only three short-axis views. In the present study, we calculate the % of LV mass covered by these three short-axis slices of T1 mapping and find an average coverage of 17 ± 4%, which is a big limitation of this technique, particularly in cardiac conditions characterized by regional or focal myocardial damage. For instance, myocardial infarction obviously has a regional distribution, which is confined to the vascularization territory of the culprit coronary artery [9]. Also, myocarditis could affect the whole myocardium, but the signs of myocardial damage may be focal or regionally distributed [8]. Thus, the three-slices approach of T1 mapping may be inaccurate and less effective than other techniques for which a complete coverage of LV is usually performed.

Recent modified Lake Louise criteria [24] included T1 and T2 mapping as diagnostic criteria and changed the original criteria by using a two-out-of-two approach: to diagnose myocarditis a “T2-based criterion”, such as edema in T2-STIR or increased native T2, should be summed to a “T1-based criterion”, such as LGE or ECV or increased native T1.

However, results of our study demonstrate that T1 mapping with the three short-axis approach is not able to detect signal abnormalities associated with myocarditis, because, in most of the cases, the signs of myocardial damage are focal. Then, the effectiveness of the new Lake Louise criteria should be assessed by further larger population studies.

The regional distribution of LGE and T2-STIR is also seen in HCM, where hypertrophy is usually asymmetric and LGE and edema are mostly located in hypertrophic segments [10]. Then, it is not surprising that we find a significant difference in the prevalence of positive LGE and/or T2-STIR and abnormal T1 in HCM.

The main results of the present study are that T1 mapping has an additive diagnostic role in only 12% of patients and is ineffective in 24% of them. To interpret these results, other limitations of T1 mapping should be considered. Indeed, the T1 mapping technique is far from being standardized, because it is well-known that differences in magnetic field shimming, pulse sequence and parameters, patient’s heart rate, and in the algorithm of the post-processing software could modify the result of this technique.

Our results suggest that a complete CMR protocol cannot be exempt anymore from T1 mapping techniques. However, T1 mapping cannot substitute conventional approaches based on LGE and T2-STIR techniques. T1 mapping could be used as a substitute for LGE only in the presence of contraindication for contrast media.

### Limitations

T1 mapping was acquired using only three short-axis slices and, as discussed above, this approach intrinsically limited the effectiveness of T1 mapping to identify focal and/or regional myocardial tissue abnormalities. However, this was indicated by the most recent SCMR consensus document [8].

We did not include T2, T2*, and ECV mapping in the present study. T2 mapping was not available in our laboratory at the time we started the enrolment and in order to have a homogeneous population, we decided not to include T2 mapping data in the present study. T2* mapping was used in selected indications such as in cardiac hemochromatosis or in the suspicion of hemorrhagic infarction, and we followed these indications and did not acquire T2* in all the patients. Finally, ECV mapping required hematocrit obtained the day of CMR, and we had this in only a minority of patients (60 patients) and we preferred not to include these data.

Finally, cardiac tumors and congenital heart disease, which are frequent indications for CMR, were not included in our population. However, we aimed to evaluate the effectiveness of T1 mapping in different cardiac diseases based on previous evidence. In contrast, the role of T1 mapping for the evaluation of cardiac tumors and congenital disease is still under evaluation.

## 5. Conclusions

Conventional CMR techniques and T1 mapping are complementary. Globally, T1 mapping may give additive information in 12% of patients but cannot substitute conventional techniques, because it is less effective particularly in conditions presenting with regional or segmental distribution of myocardial damage. This limitation could be overcome by covering the entire myocardium with T1 mapping instead of a three short-axis approach. However, T1 mapping is more effective than conventional techniques in cardiac conditions with diffuse myocardial damage, such as in the early stages of scleroderma, Fabry disease, and cardiac amyloidosis. Results of this study suggest that both conventional and mapping techniques should be acquired in all the patients but with a substantial increase in exam duration.

## Figures and Tables

**Figure 1 diagnostics-13-02461-f001:**
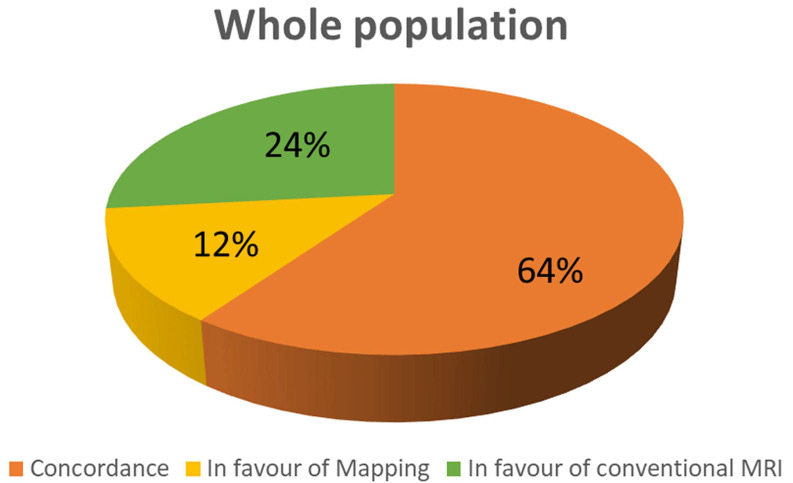
Concordance between conventional technique and T1 mapping in the whole population.

**Figure 2 diagnostics-13-02461-f002:**
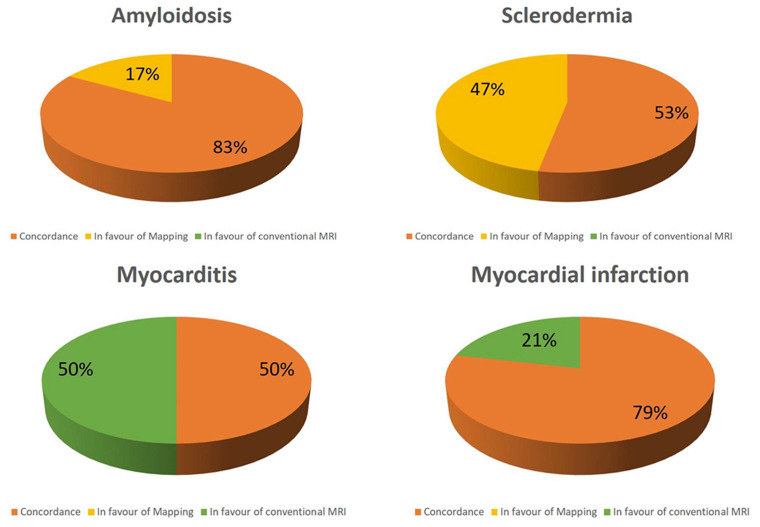
Concordance between conventional technique and T1 mapping in different subgroups.

**Figure 3 diagnostics-13-02461-f003:**
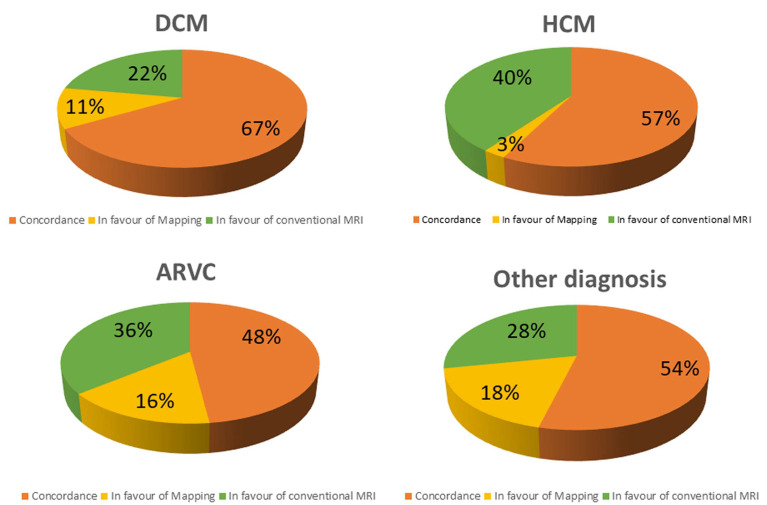
Concordance between conventional technique and T1 mapping in different subgroups (DCM, dilated cardiomyopathy; HCM, hypertrophic cardiomyopathy; ARVC, arrhythmogenic right ventricular cardiomyopathy).

**Figure 4 diagnostics-13-02461-f004:**
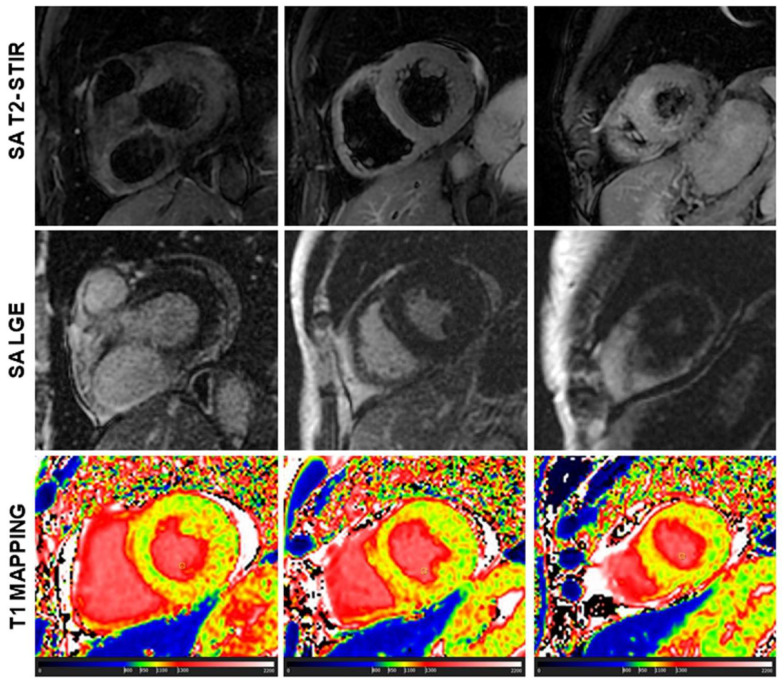
A case of scleroderma with negative late gadolinium enhancement (LGE) and T2-weighted image (T2-STIR, short tau inversion recovery) but with a diffuse increase in myocardial native T1 (normal range of T1 in green).

**Figure 5 diagnostics-13-02461-f005:**
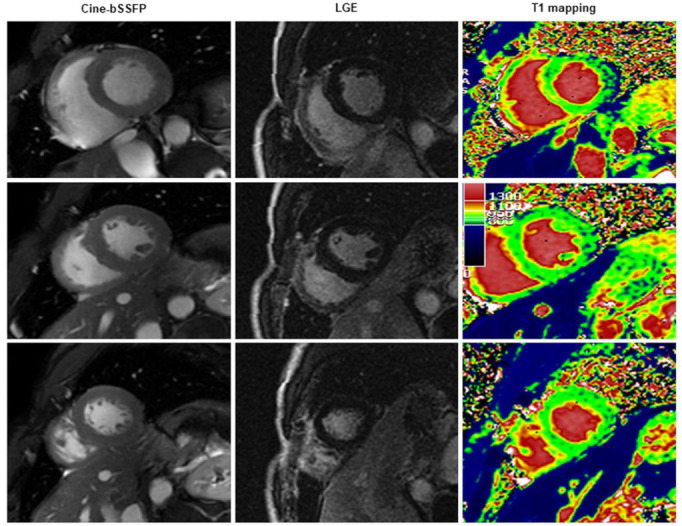
A case of Fabry disease with mild concentric hypertrophy and low T1 (blue region) with negative conventional techniques.

**Table 1 diagnostics-13-02461-t001:** Clinical indications and diagnostic features (part 1).

Clinical Indication	Morphologic Features (Cine-SSFP)	Edema/Fat	LGE	Native T1
DCM	LV dilation and dysfunction	Non-specific pattern	- Absent (75%) - Non-ischemic (20–25%)	Slight diffuse increase
HCM	Asymmetrical hypertrophy (septal, apical, septal–apical, diffuse, lateral) Secondary features (intramyocardial coronary artery bridge, apical aneurism, crypts, mitral anterior leaflet elongation, papillary muscles abnormalities)	Mid-wall edema in hypertrophied segments associated with LGE (40%)	Mid-wall distribution in hypertrophied segments (55–95%)	Focal increase on T1 only in scar region (55–95%)
ARC	- Regional wall motion abnormalities and/or dilation and/or dysfunction - of RV in RV presentation or in biventricular presentation - of LV in LV-dominant presentation - RV or LV intramyocardial India ink in fat infiltration	Fat infiltration/metaplasia	Transmural of RV walls (20–60%) Non-ischemic in LV walls (50%)	Increased in scar region Decreased in fat infiltration
Myocarditis	- None - Regional wall motion abnormalities (in coronary territory) - Global LV dysfunction	Non-ischemic pattern (85–100%)	Non-ischemic pattern (85–100%)	Increased
MINOCA	- Regional wall motion abnormalities - Potentially LV dysfunction	Ischemic-like pattern (100% in acute phase)	Ischemic presentation with potential no-reflow (hypointensity within hyperintense regions) (100%)	- Increased - Decrease in presence of haemorrhagic infarction
Tako-tsubo	Regional wall motion abnormalities in apical segments (apical ballooning)	Transmural in apical regions (100% in acute phase)	Absent	Increased in apical regions

LGE, late gadolinium enhancement; DCM, dilated cardiomyopathy; HCM, hypertrophic cardiomyopathy; ARC, arrhythmogenic cardiomyopathy; MINOCA, myocardial infarction with non-obstructed coronary arteries.

**Table 2 diagnostics-13-02461-t002:** Clinical indications and diagnostic features (part 2).

Clinical Indication	Morphologic Features (Cine SSFP)	Edema	LGE	Native T1
Scleroderma	Non-specific	Diffuse/focal edema in active inflammation (possible)	- Normal - Diffuse mild-enhanced - Focal hyper-enhanced LGE (% unknown)	Diffuse/focal increase
Amyloidosis	- Concentric hypertrophy - Left atrial wall thickening - Pericardial effusion	Non-specific pattern	Specific pattern: - Diffuse subendocardial pattern - Nulling defect of myocardium - Early post-contrast darkening of blood	Diffuse increase
Myocardial infarction	- Regional wall motion abnormalities (in coronary territory) - Potentially LV dysfunction	- Ischemic-like pattern in acute (100%) - Absent in chronic	Ischemic presentation In acute setting with potential no-reflow (hypointensity within hyperintense regions) (100%)	- Increased in scar and acute infarction - Decrease in presence of haemorrhagic infarction
Fabry disease	- Concentric hypertrophy (90%) - Asymmetric hypertrophy (10%)	Non-specific pattern	Mid-wall inferolateral in late stages (40%) Mid-wall septal or apical (4%)	- Diffusely decreased - In late stage with extensive LGE, T1 may increase (pseudonormalization)
Dystrophy	- Normal - LV dysfunction - Regional wall motion abnormalities - India ink	Non-specific pattern	Non-ischemic pattern (10–20%)	- Increased in scar region - Decreased in fat infiltration
Pericarditis	- Pericardial effusion - Thickening of pericardial layers	Hyperintensity of pericardial layers in pericarditis (80%)	- Enhancement of layer in pericarditis - No enhancement in non-inflammatory effusion (80%)	Unknown

LGE, late gadolinium enhancement.

**Table 3 diagnostics-13-02461-t003:** Functional characteristics.

Indication	*n*	Males	Age	LV EF	LV EDVi	RV EF	RV EDVi
DCM	61 (18.9)	47 (77)	59 ± 14	43.15 ± 14.58	112.56 ± 35.31	56.16 ± 11.54	78.21 ± 21.39
HCM	51 (15.8)	35 (69)	60 ± 13	67.43 ± 11.31	73.80 ± 21.64	65.29 ± 7.94	69.59 ± 21.74
ARC	63 (19.5)	43 (68)	40 ± 17	65.08 ± 7.64	87.79 ± 16.22	60.74 ± 7.38	91.79 ± 19.85
Myocarditis	44 (13.6)	29 (66)	50 ± 20	61.34 ± 9.04	77.86 ± 18.39	61.61 ± 5.93	76.09 ± 17.55
Scleroderma	21 (6.5)	2 (10)	56 ± 15	61.29 ± 16.71	76.81 ± 16.97	62.38 ± 7.51	75.19 ± 16.48
Amyloidosis	17 (5.3)	9 (53)	76 ± 8	63.06 ± 14.25	70.94 ± 20.64	66.76 ± 9.48	59.94 ± 12.43
Myocardial infarction (acute/chronic)	15 (4.6)	11 (73)	62 ± 15	49.73 ± 16.42	89.60 ± 27.13	63.80 ± 8.17	61.47 ± 14.73
Dystrophy/mitochondrial	8 (2.8)	7 (85)	41 ± 17	62.00 ± 9.51	70.67 ± 20.56	62.78 ± 9.68	65.44 ± 15.62
Pericarditis–pericardial effusion	6 (1.9)	2 (33)	46 ± 13	66.50 ± 10.21	80.50 ± 15.14	66.67 ± 6.95	71.83 ± 13.06
LV non-compaction	4 (1.2)	2 (50)	39 ± 21	65.50 ± 10.63	83.25 ± 11.21	58.75 ± 7.41	87.75 ± 16.32
Systemic sarcoidosis	3 (0.9)	1 (33)	56 ± 9	46.67 ± 21.01	74.00 ± 3.61	38.00 ± 22.61	97.00 ± 32.51
Fabry	1 (0.3)	1 (100)	45	65	70	63	72
Pulmonary hypertension	1 (0.3)	1 (100)	58	57	83	49	81
Valvular disease	1 (0.3)	1 (100)	77	58	88	70	78
Healthy controls	27 (8.4)	16 (59)	51 ± 18	67.15 ± 6.73	78.44 ± 18.25	63.59 ± 7.00	78.07 ± 19.48

DCM, dilated cardiomyopathy; HCM, hypertrophic cardiomyopathy; ARC, arrhythmogenic LV, left ventricle; RV, left ventricle; EDVi, end-diastolic volume index; EF, ejection fraction.

**Table 4 diagnostics-13-02461-t004:** CMR findings of conventional sequences and T1 mapping.

Indication	T2-STIR+	LGE+	T1 Regional Abnormality	Mean T1
DCM	1 (1.6)	44 (73.3)	37 (60.7%)	1050 ± 44
HCM	2 (3.9)	37 (74)	19 (37.3%)	1029 ± 48
ARVC	2 (3.2)	31 (50.8)	19 (30.2%)	1012 ± 54
Myocarditis	27 (61.4)	33 (75.0)	25 (56.8%)	1053 ± 65
Scleroderma	4 (19.0)	10 (50.0)	18 (85.7%)	1092 ± 41
Amyloidosis	0	12 (70.6)	11 (64.7%)	1063 ± 78
Myocardial infarction (acute/chronic)	3 (20.0)	15 (100)	11 (73.3%)	1063 ± 87
Dystrophy/mitochondrial disease	0 (0.0)	6 (66.7)	6 (66.7%)	1058 ± 65
Pericarditis–pericardial effusion	1 (16.7)	1 (16.7)	3 (50.0%)	1105 ± 102
LV non-compaction	0 (0.0)	4 (100)	2 (50.0%)	1059 ± 76
Systemic sarcoidosis	1 (33.3)	2 (66.7)	2 (66.7%)	1049 ± 33
Fabry	0	1 (100)	1 (100)	827
Pulmonary hypertension	0	1 (100)	0 (0.0%)	1026
Valvular disease	0	1 (100)	0 (0.0%)	1018
Healthy controls	0 (0.0)	0 (0.0	0(0.0%)	1034 ± 29

DCM, dilated cardiomyopathy; HCM, Hypertrophic cardiomyopathy; ARVC, arrhythmogenic cardiomyopathy, LV, left ventricle. STIR, short tau inversion recovery.

**Table 5 diagnostics-13-02461-t005:** CMR findings divided by the initial suspicion.

Initial Suspicion	Specific Findings *N* (%)	Alternative Diagnosis *N* (%)	Non-Specific Findings *N* (%)	Negative CMR *N* (%)
DCM	34 (55.7)	7 (11.5)	17 (27.9)	3 (4.9)
HCM	33 (64.7)	2 (3.9)	16 (31,4)	0
ARVC	12 (19.0)	11 (17.5)	22 (34.9)	18 (28.6)
Myocarditis	22 (50)	14 (31.8)	6 (13.6)	2 (4.6)
Scleroderma	17 (80.9)	0	3 (14.3)	1 (4.8)
Amyloidosis	5 (29.4)	4 (23.5)	6 (35.3)	2 (11.8)
Myocardial infarction (acute/chronic)	10 (66.7)	1 (6.6)	4 (26.7)	0
Dystrophy/mitochondrial disease	8 (88.9)	0	0	1 (11.1)
Pericarditis–pericardial effusion	4 (66.6)	1 (16.7)	1 (16,7)	0
LV non-compaction	1 (25)	1 (25)	2 (50)	0
Systemic sarcoidosis	1 (33.3)	0	1 (33.3)	1 (33.3)
Fabry	1(100)	0	0	0
Pulmonary hypertension	1 (100)	0	0	0
Valvular disease	1 (100)	0	0	0
Total	149 (50.3)	41 (13.8)	78 (26.4)	28 (9.5)

DCM, dilated cardiomyopathy; HCM, hypertrophic cardiomyopathy; ARVC, arrhythmogenic cardiomyopathy.

## Data Availability

The data presented in this study are available on request from the corresponding author.

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
