# Peer review of "Diagnostic Role of Native T1 Mapping Compared to Conventional Magnetic Resonance Techniques in Cardiac Disease in a Real-Life Cohort"

_diagnostics, 2023, doi:10.3390/diagnostics13142461_

Round 1
Reviewer 1 Report
1. Figures: it is needed to put the abreviations
2. in the figures of CMR please put The ROI where you have measured the mapping
3.please detail which are the normal values for your machine, a 50 patients cohort is needed in this case,
4. did you measured ECV?
Author Response
Dear Reviewer,
many thanks for you revision.
Answer to specific comments:
1)Figures: it is needed to put the abreviations
Answer: we included abbreviation in figure legends
2)in the figures of CMR please put The ROI where you have measured the mapping
Answer: we used circle CVI42 for the analysis that permitted to have a measurement of the 16 AHA\ACC\ESC wall segments. A ROI was occasionally placed to detect focal involvement which is not the case of the 2 examples included.
3)please detail which are the normal values for your machine, a 50 patients cohort is needed in this case
Answer: We used our normality range of our institution performed in 100 sex- and age- group of patients and pubblished in 2021
(Meloni A, Martini N, Positano V, D'Angelo G, Barison A, Todiere G, Grigoratos C, Barra V, Pistoia L, Gargani L, Ripoli A, Pepe A. Myocardial T1 Values at 1.5 T: Normal Values for General Electric Scanners and Sex-Related Differences. J Magn Reson Imaging. 2021 Nov;54(5):1486-1500.)
4)did you measured ECV?
Answer: As already specify in the limitation section of discussion "ECV mapping required hematocrit obtained the day of CMR, we had this in only a minority of patients (60 patients) and we preferred not to include that data."
Reviewer 2 Report
The paper presents Diagnostic Role of Native T1 mapping Compared to Conventional Magnetic Resonance Techniques in Cardiac Disease in a Real-life Cohort. Overall, the scientific objective is important.
The article is well written and comprehensive.
There are no ethical concerns about this study.
The research design is appropriate and the methods clearly explained.
The interpretation of the results is clearly presented and adequately supported by the evidence adduced.
The references are up-to-date and the most important studies have been cited.
There are some minor/major revisions needed. Please provide a point-by-point response to the following queries.
1. Figure 4 and 5- Images are of poor quality and unreadable, please correct that
2. Please explain the abbreviations used in the figures and tables in their footnotes.
3.The methodology section should provide information on inclusion and exclusion criteria.
4. The results should information on numbers of patients screened, with reasons for exclusions at each stage.
Author Response
Dear Reviewer,
many thanks for the recognition of the quality of our work and of your comments.
Answer of specific comments:
1)Figure 4 and 5- Images are of poor quality and unreadable, please correct that
Answer: Actually images were in tiff format with 300 DPI and 1884x1635 pixels. In the word file, the quality depends on the word processor softwatre used and if a memory saving option is chosen. Have you read the manuscript using a tablet? or with a "preview" mode? In microsoft word the image are quite good. In the final pdf will be good.
2)Please explain the abbreviations used in the figures and tables in their footnotes.
Answer: As suggested we spelled out the abbreviatons
3)The methodology section should provide information on inclusion and exclusion criteria.
Answer: As suggested we provide such information: "We excluded patients with congenital heart disease (T1 mapping was not included in clinical protocol) and those with cardiac tumors. Finally, after CMR patients with sub-optimal images, those with contraindication for gadolinium-based agents and those that did not complete the CMR exam were excluded."
4)The results should information onnumbers of patients screened, with reasons for exclusions at each stage.
Answer: Among 340 pts enrolled, the final population included 323 pts, The final population included 323 patients (17 excluded for sub-optimal image quality or for incomplete exam). This information is included in the method section.
Round 2
Reviewer 1 Report
All changes in the paper have been done, can be published.